# Interaction of Subculture Cycle, Hormone Ratio, and Carbon Source Regulates Embryonic Differentiation of Somatic Cells in *Pinus koraiensis*

Yuhui Ren [1,†], Xiaoqian Yu [1,†], Honglin Xing [1], Iraida Nikolaevna Tretyakova [2], Alexander Mikhaylovich Nosov [3,4], Ling Yang [1,5,*] and Hailong Shen [1,5,*]

1   State Key Laboratory of Tree Genetics and Breeding, School of Forestry, Northeast Forestry University, Harbin 150040, China
2   Laboratory of Forest Genetics and Breeding, Institution of the Russian Academy of Sciences V.N. Sukachev Institute of Forest Siberian Branch of RAS, Krasnoyarsk 660036, Russia
3   Department of Cell Biology, Institute of Plant Physiology K.A. Timiryazev Russian Academy of Sciences, Moscow 127276, Russia
4   Department of Plant Physiology, Biological Faculty, Lomonosov Moscow State University, Moscow 119991, Russia
5   State Forestry and Grassland Administration Engineering Technology Research Center of Korean Pine, Harbin 150040, China
*   Correspondence: yangl-cf@nefu.edu.cn (L.Y.); shenhl-cf@nefu.edu.cn (H.S.)
†   These authors contributed equally to this work.

**Abstract:** During somatic embryogenesis of *Pinus koraiensis*, the ability of the embryogenic callus to produce embryos gradually decreases with long-term proliferative culture, which seriously affects large-scale application of this technology. In this study, embryogenic calli of Korean pine 1–100 cell line were used as materials. It was found that in the 7-day subculture cycle of embryogenic calli the somatic embryogenic potential of Korean pine remained for the longer time. In comparison with a subculture cycle of 14 days, indoleacetic acid (IAA), soluble sugar, and starch contents in embryogenic callus were higher with a subculture cycle of 7 days, while hydrogen peroxide ($H_2O_2$) content showed the opposite trend. Further, low levels of endogenous abscisic acid (ABA) and high levels of IAA could inhibit somatic embryogenesis of *P. koraiensis* calli. Among different carbon sources, maltose produced more storage materials and higher superoxide dismutase (SOD) and catalase (CAT) activities than sucrose, which was conducive to cell differentiation and somatic embryogenesis. The results revealed the physiological characteristics of the loss of embryogenic capacity of Korean pine embryogenic callus and provided a basis for the long-term maintenance of its somatic embryogenic capacity.

**Keywords:** carbon source; embryogenic callus; hormone ratio; Korean pine; subculture cycle; somatic embryogenesis

## 1. Introduction

Somatic embryogenesis is an important system of plant somatic genesis and genetic transformation, which has the potential of large-scale seedling breeding [1]. Somatic embryogenesis decreases with long-term proliferative culture for most species of conifer [2]. This phenomenon has seriously hindered the application of somatic embryogenesis in high-efficiency breeding. From studies on long-term subculture embryogenic maintenance, a suitable subculture cycle can preserve good growth of embryogenic callus [3–5]. For most plants, 2.4-dichlorophenoxy acetic acid (2, 4−D) is the auxin used to induce somatic embryos. In the proliferation stage, with the passage of time, the concentration of 2, 4−D should be reduced or replaced with NAA, which is not only conducive to the proliferation

of embryonic callus, but also conducive to the transformation of more embryonic cell clusters into embryonic suspension cell clusters in the later stage [6–8]. In addition, sucrose is the most commonly used carbon source and osmotic regulator [9]. The addition of sucrose to the culture medium fosters the accumulation of dry matter in somatic embryos and plays an important role in promoting somatic embryogenesis, in particular the formation of cotyledon embryos [10]. For somatic embryogenesis, maltose was found to be the best carbon source for maintaining embryogenic callus in *Pinus pinaster* [3] and *Pinus taeda* [11]. However, to date, research on the factors influencing conifer somatic embryogenesis and development has been incomplete, and there is no general model for the study of maintaining somatic embryogenesis of different tree species.

*P. koraiensis*, a member of the Pinaceae family, is a dominant species of the broad-leaved *P. koraiensis* forest in Northeast China [12]. The traditional propagation methods of *P. koraiensis* include sowing and grafting, but both have disadvantages of low reproduction coefficient and large genetic variation [13]. During somatic embryogenesis of *P. koraiensis*, the ability of embryogenic callus to produce embryos decreases gradually with the increased duration of subculturing, and the capacity for embryonic differentiation is lost after twelve months. This phenomenon leads to the proliferation and differentiation of embryogenic callus of *P. koraiensis* being limited by the age of callus, which affects the efficiency of somatic embryo production. Therefore, the establishment of a technical system to maintain the somatic embryogenic ability of embryogenic callus of *P. koraiensis* is needed urgently.

In this study, calli induced from immature *P. koraiensis* embryos were used. Different subculture cycles, hormone ratios, and carbon sources were investigated during the subculture of embryogenic callus to maintain somatic embryogenic ability of *P. koraiensis*. In addition, morphology and physiological indices were observed to reveal the mechanism of the loss of somatic embryogenic ability of *P. koraiensis* embryogenic callus. During the propagation of *P. koraiensis*, the maintenance of somatic embryogenic ability eliminates repeated explant sampling and primary culturing and effectively improves the production efficiency of seedlings from tissue culture. In addition, long-term subculture of embryogenic callus of *P. koraiensis* and maintaining embryogenesis enables production of more somatic embryos and more regenerated plants. Further, this study provides a model for the maintenance of somatic embryogenesis of calli of other coniferous species.

## 2. Materials and Methods

### 2.1. Plant Materials

Immature cones were collected from adult mother trees in the Sanchezi forest farm, Baishan City, Jilin Province, China in July 2019. Embryogenic calli were induced from the *P. koraiensis* 1–100 cell line and were subcultured for three months [14].

### 2.2. SE Proliferation, Maturation and Capability Evaluation

The calli were stored in liquid nitrogen using cryopreservation technology. In December 2020, the embryogenic calli were removed from liquid nitrogen, resuscitated, and transferred to the proliferation medium. The proliferation medium was slightly adjusted based on Peng et al. [13]: modified Litvay medium (MLV) [15] + 2 mg·L$^{-1}$ 2,4−D + 0.5 mg·L-1 6-benzyl amino-purine (6-BA) with the addition of 25 g·L$^{-1}$ sucrose, 0.5 g·L$^{-1}$ acid hydrolyzed complex protein, 0.5 g$^{-1}$ L-glutamine, and 4 g·L$^{-1}$ gellan gum. The pH of the medium was adjusted to 5.8, then it was autoclaved at 121 °C for 30 min. Cultures were grown in the dark at 25 ± 2 °C. Maturation medium was based on Klimaszewska's formula: MLV [15] + 80 μmol·L-1 abscisic acid (ABA) + 1.2% gellan gum + 0.2 mol·L$^{-1}$ sucrose, adjusted to pH = 5.8. Each treatment was replicated 5 times. After culturing in the dark for 60 days, the number of somatic embryogenic calli per gram of embryogenic callus was counted.

*2.3. Different Subculture Cycles*

The effect of a cell cycle of 7 or 14 days on embryogenic maintenance of long-term subcultured embryogenic calli of embryogenic cell line 1–100 was investigated. Concurrently, embryogenic calli were tested for somatic maturation at the 4th, 6th, 8th, 10th, and 12th month after subculture, and physiological and biochemical indices were measured.

*2.4. Different Hormone Ratios*

The effect of hormone ratio (hormone ratios in proliferation medium: 2 mg·$L^{-1}$ 2,4−D + 0.5 mg·$L^{-1}$ 6−BA or 1 mg·$L^{-1}$ 1-naphthaleneacetic acid (NAA) + 0.25 mg·$L^{-1}$ 6−BA) on embryogenic maintenance of long-term subcultured embryogenic calli of embryogenic cell line 1–100 was explored. Concurrently, the embryogenic calli were tested for somatic maturation, and physiological and biochemical indices were measured at the 4th, 6th, 8th, 10th, and 12th month after subculture.

*2.5. Different Carbon Sources Set*

The effect of carbon source in the medium at a concentration of 0.088 mol·$L^{-1}$ (carbon source in proliferation medium: sucrose or maltose) on embryogenic maintenance of long-term subcultured embryogenic callus of embryogenic cell line 1–100 was determined. Concurrently, the embryogenic calli were tested for somatic maturation, and physiologaical and biochemical indices were measured at the 4th, 6th, 8th, 10th, and 12th month after subculture.

*2.6. Chemical Analysis of Long-Term Subcultured Embryogenic Calli*

The calli from the different treatments at the 4th, 6th, 8th, 10th, and 12th month after subculture were treated as follows:

(1) Embryogenic calli collected from the periphery of embryogenic tissue were cultured for somatic embryo maturation. A 50-mg sample of embryogenic calli was weighed, shaken in 3 mL of liquid basic medium without growth regulator and L-glutamine, and then transferred to a Buchner funnel containing sterile filter paper. The liquid medium was extracted by vacuum pump, and the filter paper containing embryogenic calli was placed on the maturation medium.

(2) Determination of the contents of the endogenous hormones ABA and indoleacetic acid (IAA): approximately 0.1 g of embryogenic callus was placed in a 1.8-mL freezing tube and stored in a −80 °C freezer. Each treatment was sampled three times. After all samples were collected and frozen in dry ice, they were sent to Shanghai Enzyme Linked Biotechnology Co., Ltd., (Shanghai China) for analysis, and the contents of ABA and IAA were determined by an enzyme linked immunosorbent assay (ELISA).

(3) Biochemical analysis of embryogenic callus: approximately 0.5 g embryogenic callus was used in each of these analyses. Corresponding assay kits purchased from Suzhou Keming Biological Co., Ltd., (Suzhou, China) were used for determination of the content of soluble protein, soluble sugar, starch, and hydrogen peroxide ($H_2O_2$), as well as the activity of peroxidase (POD), superoxide dismutase (SOD), and catalase (CAT), respectively.

*2.7. Data Analysis*

Statistical analyses of the data were conducted using Microsoft Excel 2003; multiple comparisons (Duncan's multiple range test) and analysis of variance were performed using SPSS (2010, V. 19.0: SPSS, Inc., Cary, NC, USA). Significant differences were determined at the level of $p = 0.05$. Finally, SigmaPlot (v12.5, SYSTAT, San Jose, CA, USA) was used for generating figures.

## 3. Results

*3.1. Effects of Different Culture Methods on the Number of Somatic Embryos*

Overall, the number of somatic embryos produced from a subculture cycle of 7 days initially increased then decreased. The number of somatic embryos at the 4th and 6th

months of subculture was lower for 7 days than for 14 days. From the 8th month of subculture, the number of somatic embryos produced from a subculture cycle of 7 days was higher than that from 14 days. Embryogenic calli decreased overall with the extension of subculture time, and the number of somatic embryos was 0 in the 12th month of subculture, indicating that their somatic embryogenic ability had been lost entirely (Figure 1a). On the media supplemented with 2 mg·L$^{-1}$ 2, 4−D + 0.5 mg·L$^{-1}$ 6−BA and 1 mg·L$^{-1}$ NAA + 0.25 mg·L$^{-1}$ 6−BA, the number of somatic embryos in the 4th month of subculture reached the maximum values of 118·g$^{-1}$ fresh weight (FW) and 126·g$^{-1}$ FW, respectively. Apart from the 6th month of subculture, the number of basal body embryos cultured with 1 mg·L$^{-1}$ NAA + 0.25 mg·L$^{-1}$ 6−BA was higher than with 2 mg·L$^{-1}$ 2, 4−D + 0.5 mg·L$^{-1}$ 6−BA (Figure 1b).

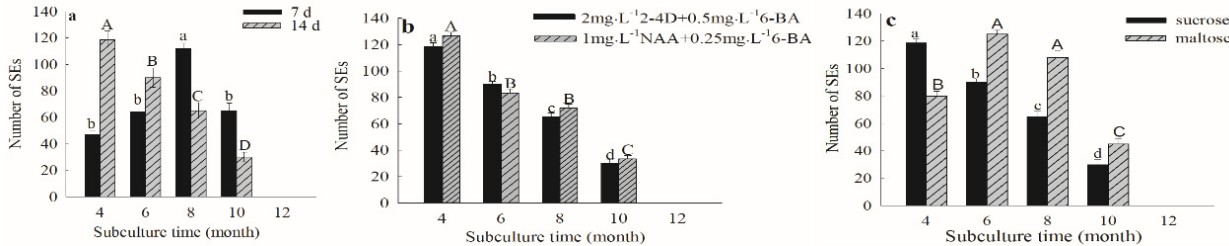

**Figure 1.** Effects of subculture cycle, hormone ratio, and carbon source on the number of somatic embryos in *Pinus koraiensis*. (**a**) Number of somatic embryos in different subculture cycles, (**b**) Number of somatic embryos with different hormone ratios, (**c**) Number of somatic embryos with different carbon source typest; Note: Different letters (upper or lower case) in the table indicate significant differences between data columns (Duncan's multiple range test; *p* = 0.05).

In the subculture medium with sucrose as the carbon source, the number of somatic embryos decreased overall and reached the maximum value (118·g$^{-1}$ FW) in the 4th month. In the subculture medium with maltose as the carbon source, the number of somatic embryos initially increased and then decreased with increased subculture time. The number of somatic embryos reached the maximum (125·g$^{-1}$ FW) in the 6th month of subculture. The number of somatic embryos in the medium with maltose as the carbon source was higher than that in the medium with sucrose as the carbon source (Figure 1c).

### 3.2. Effects of Different Culture Methods on Endogenous Hormones

#### 3.2.1. ABA Content Change

ABA content in embryogenic callus with a subculture cycle of 7 days reached the maximum (462 ng·g$^{-1}$) in the 8th month of subculture. ABA content in embryogenic callus with a subculture cycle of 14 days decreased with the extension of subculture time, reaching the minimum value (448.13 ng·g$^{-1}$) in the 12th month (Figure 2a). ABA content of embryogenic callus on the medium supplemented with 2 mg·L$^{-1}$ 2, 4−D + 0.5 mg·L$^{-1}$ 6−BA decreased gradually with increased subculture time. ABA content on the medium supplemented with 1 mg·L$^{-1}$ NAA + 0.25 mg·L$^{-1}$ 6−BA initially increased and then decreased with the extension of subculture time. ABA content reached the maximum value (525.57 ng·g$^{-1}$) in the eighth month of subculture (Figure 2b). On the medium with sucrose as the carbon source, ABA content in the 4th month of subculture reached a maximum of 550.57 ng·g$^{-1}$, which was 102.44 ng·g$^{-1}$ higher than in the 12th month of subculture. On the medium with maltose as the carbon source, ABA content of embryogenic callus increased initially and then decreased consistently, and ABA content reached the maximum (510.93 ng·g$^{-1}$) in the 8th month of subculture (Figure 2c).

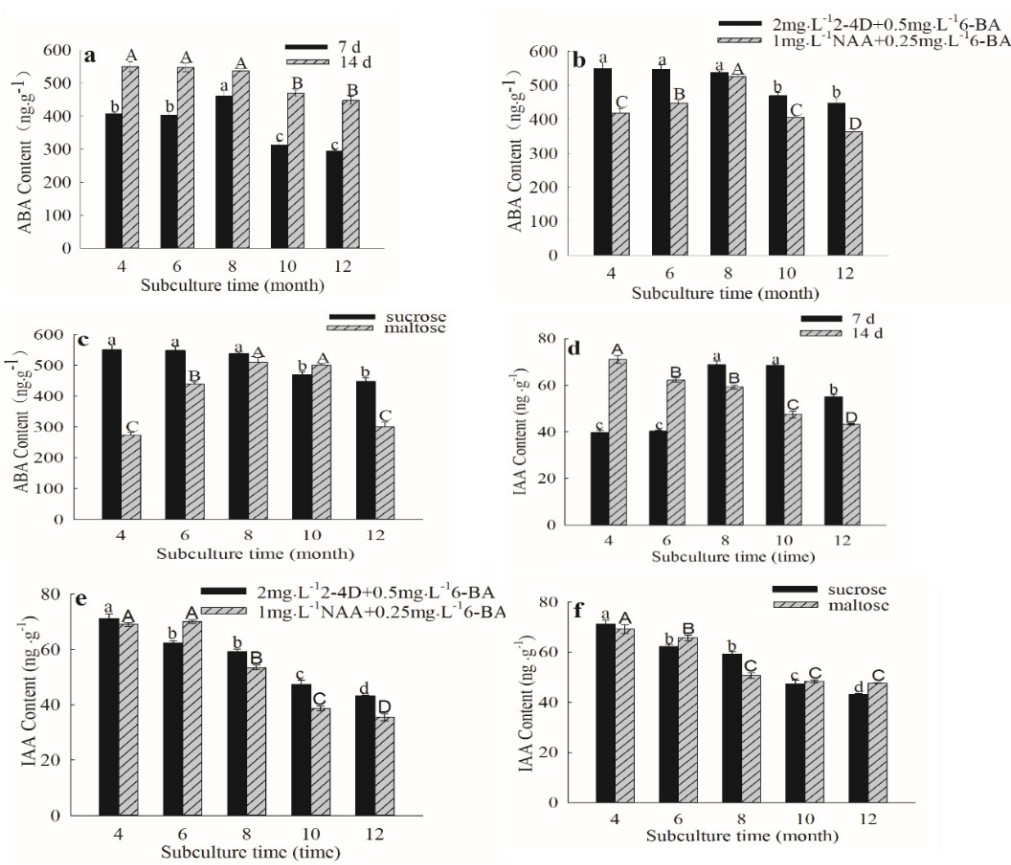

**Figure 2.** Effects of subculture cycle, hormone ratio, and carbon source on ABA and IAA contents of *Pinus koraiensis* embryogenic callus during long-term subculture. (**a**) ABA content of callus in different subculture cycles, (**b**) ABA content of callus with different hormone ratios, (**c**) ABA content in callus of different carbon sources, (**d**) IAA content of callus in different subculture cycles, (**e**) IAA content of callus with different hormone ratios, (**f**) IAA content in callus of different carbon sources; Note: Different letters (upper or lower case) in the table indicate significant differences between columns (Duncan's multiple range test; *p* = 0.05).

### 3.2.2. IAA Content Change

Overall, IAA content in embryogenic callus with a subculture cycle of 7 days increased initially and then decreased with the extension of subculture time, reaching the maximum value (68.89 ng·g$^{-1}$) in the 8th month of subculture. IAA content in embryogenic callus with a subculture cycle of 14 days decreased overall with the extension of subculture time. IAA content in the 12th month of subculture was 27.88 ng·g$^{-1}$ lower than in the 4th month (Figure 2d). In the media supplemented with 2 mg·L$^{-1}$ 2,4−D + 0.5 mg·L$^{-1}$ 6−BA and 1 mg·L$^{-1}$ NAA + 0.25 mg·L$^{-1}$ 6−BA, IAA content of embryogenic callus gradually decreased overall with the extension of subculture time, and the maximum values of 71.15 ng·g$^{-1}$ and 70.05 ng·g$^{-1}$, respectively, occurred in the 4th month of subculture (Figure 2e). IAA content of embryogenic callus in the media with sucrose and maltose as carbon sources decreased (Figure 2f).

### 3.2.3. Change in Stored Substance Content
#### Effect of Subculture Cycle on Stored Substance Content

The soluble sugar content of embryogenic calli with a subculture cycle of 7 days increased gradually as subculture time increased. From the 8th month of subculture, the soluble sugar content accumulated rapidly and reached the maximum value (3 mg·mg$^{-1}$ prot) in the 12th month. The soluble sugar content of embryogenic calli with a subculture cycle of 14 days generally decreased with increased subculture time. The soluble sugar

content reached the maximum in the 4th month of subculture (2.41 mg·mg$^{-1}$ prot), which was 2.68 times greater than in the 12th month (Figure 3a). The starch content of embryogenic calli in the two subculture cycles decreased with increased subculture time, reaching the maximum values of 2.55 mg·mg$^{-1}$ prot and 1.72 mg·mg$^{-1}$ prot in the 4th month of subculture and the minimum values of 1.16 mg·mg$^{-1}$ prot and 0.72 mg·mg$^{-1}$ prot in the 12th month of subculture for 7 and 14 days of subculture, respectively (Figure 3b). In addition, the soluble protein content of embryogenic calli in the two subculture cycles decreased gradually overall, and the difference was not significant (Figure 3c).

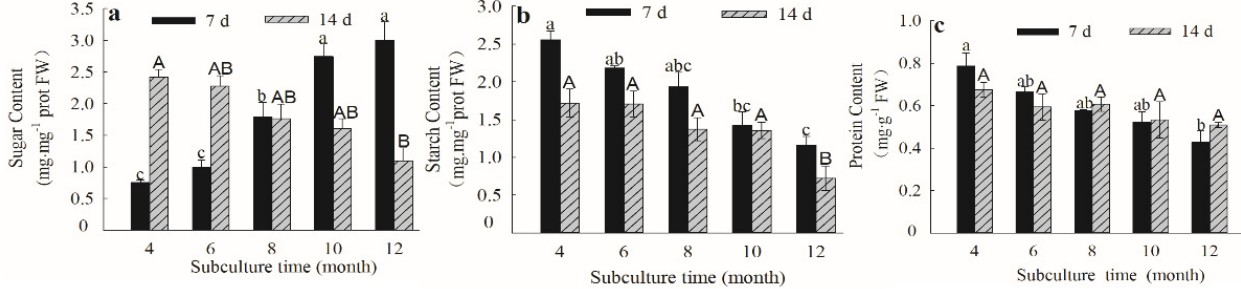

**Figure 3.** Effects of subculture cycle on storage substances in embryogenic calli of Pinus koraiensis during long-term subculture. (**a**) Sugar content of callus in different subculture cycles, (**b**) Starch content of callus in different subculture cycles, (**c**) Protein content of callus in different subculture cycles; Note: Different letters (upper or lower case) in the table indicate significant differences between columns (Duncan's multiple range test; *p* = 0.05).

Effect of hormone ratio on storage material content

The soluble sugar, starch, and protein contents of embryogenic calli were highest in the 4th month of subculture on medium supplemented with 2 mg·L$^{-1}$ 2,4−D + 0.5 mg·L$^{-1}$ 6−BA (2.41 mg·mg$^{-1}$ prot, 1.72 mg·mg$^{-1}$ prot, and 0.67 mg·mg$^{-1}$, respectively) and on medium supplemented with 1 mg·L$^{-1}$ NAA + 0.25 mg·L$^{-1}$ 6−BA (2.89 mg·mg$^{-1}$ prot, 1.94 mg·mg$^{-1}$ prot, and 0.88 mg·mg$^{-1}$, respectively). The soluble sugar, starch, and protein contents decreased to minimum values in the 12th month on medium supplemented with 2 mg·L$^{-1}$ 2,4−D + 0.5 mg·L$^{-1}$ 6−BA (1.10 mg·mg$^{-1}$ prot, 0.72 mg·mg$^{-1}$ prot, and 0.51 mg·mg$^{-1}$, respectively) and on medium supplemented with 1 mg·L$^{-1}$ NAA + 0.25 mg·L$^{-1}$ 6−BA (1.36 mg·mg$^{-1}$ prot, 1.03 mg·mg$^{-1}$ prot, and 0.33 mg·mg$^{-1}$, respectively). The soluble sugar and protein contents of embryogenic callus on the medium supplemented with 2 mg·L$^{-1}$ 2,4−D + 0.5 mg·L$^{-1}$ 6−BA were higher than those on the medium supplemented with 1 mg·L$^{-1}$ NAA + 0.25 mg·L$^{-1}$ 6−BA, but the opposite was found for starch content (Figure 4).

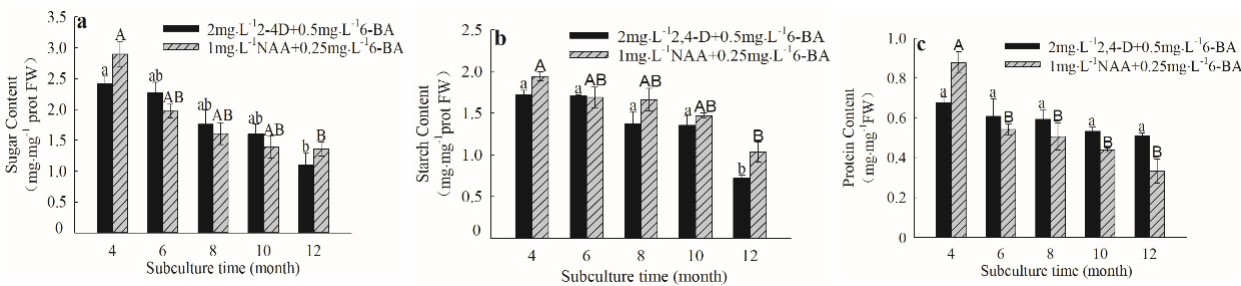

**Figure 4.** Effects of subculture cycle and hormone ratio on storage substances in embryogenic calli of Pinus koraiensis during long-term subculture. (**a**) Sugar content in callus with different hormone ratios, (**b**) Starch content in callus with different hormone ratios, (**c**) The content of callus protein with different hormone ratios; Note: Different letters (upper or lower case) in the table indicate significant differences between columns (Duncan's multiple range test; *p* = 0.05).

Effect of carbon source on stored material content

On the medium with sucrose as the carbon source, the maximum soluble sugar content of embryogenic callus was 2.41 mg·mg$^{-1}$ prot in the 4th month of subculture. On the medium with maltose as the carbon source, the soluble sugar content of embryogenic callus increased initially and then decreased with increased subculture time; the maximum value of 2.60 mg·mg$^{-1}$ prot was reached in the 8th month of subculture (Figure 5a). On the medium with sucrose as the carbon source, the starch content of embryogenic callus decreased gradually with increased subculture time. The maximum value of 1.72 mg·mg$^{-1}$ prot occurred in the 4th month of subculture and was 1.00 mg·mg$^{-1}$ prot greater than in the 12th month of subculture. On the medium with maltose as the carbon source, the starch content initially increased and then decreased with increased subculture time; the maximum value of 3.53 mg·mg$^{-1}$ prot was reached in the 8th month of subculture, which then gradually decreased to the minimum value of 1.57 mg·mg$^{-1}$ prot in the 12th month (Figure 5b). The protein content of embryogenic callus decreased gradually with increased subculture time on both the culture medium with sucrose and maltose as the carbon sources. The maximum values of 0.88 mg·g$^{-1}$ FW and 0.97 mg·g$^{-1}$ FW for sucrose and maltose, respectively, occurred in the 4th month of subculture and were 0.55 mg·g$^{-1}$ FW and 0.64 mg·g$^{-1}$ FW, respectively, higher than those in the 12th month of subculture (Figure 5c).

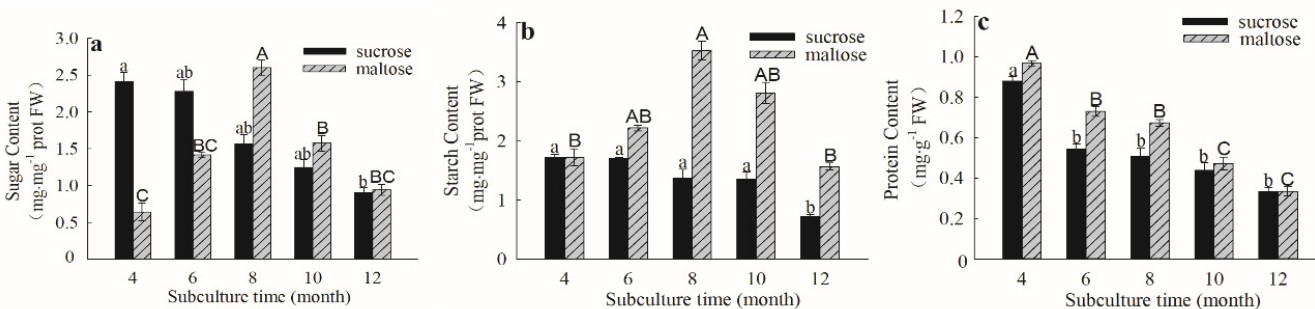

**Figure 5.** Effects of carbon sources on storage material contents of embryogenic callus of *Pinus koraiensis* during long-term subculture. (**a**) Sugar content in callus with different carbon sources, (**b**) Starch content in callus with different carbon sources, (**c**) The content of callus protein with different carbon sources; Note: Different letters (upper or lower case) in the table indicate significant differences between columns (Duncan's multiple range test; *p* = 0.05).

### 3.2.4. Antioxidants

Effect of Subculture Cycle on Antioxidants

Prolonging the time of subculture led to decreasing SOD activity of embryogenic callus in advance of reaching the maximum. The highest SOD activities, 1674.41 U·mg$^{-1}$ prot and 1126.53 U·mg$^{-1}$ prot, were reached in the 8th month and in the 4th month for subculture cycles of 7 days and 14 days of embryogenic callus, respectively (Figure 6a). In addition, the changes of CAT activity, POD activity, and $H_2O_2$ content of somatic embryos in the 7-day and 14-day subculture cycles were consistent, reaching the maximum values in the 8th month (Figure 6b and Figure S1a, online Supplementary Data) and the 12th month (Figure S1b, online Supplementary Data), respectively.

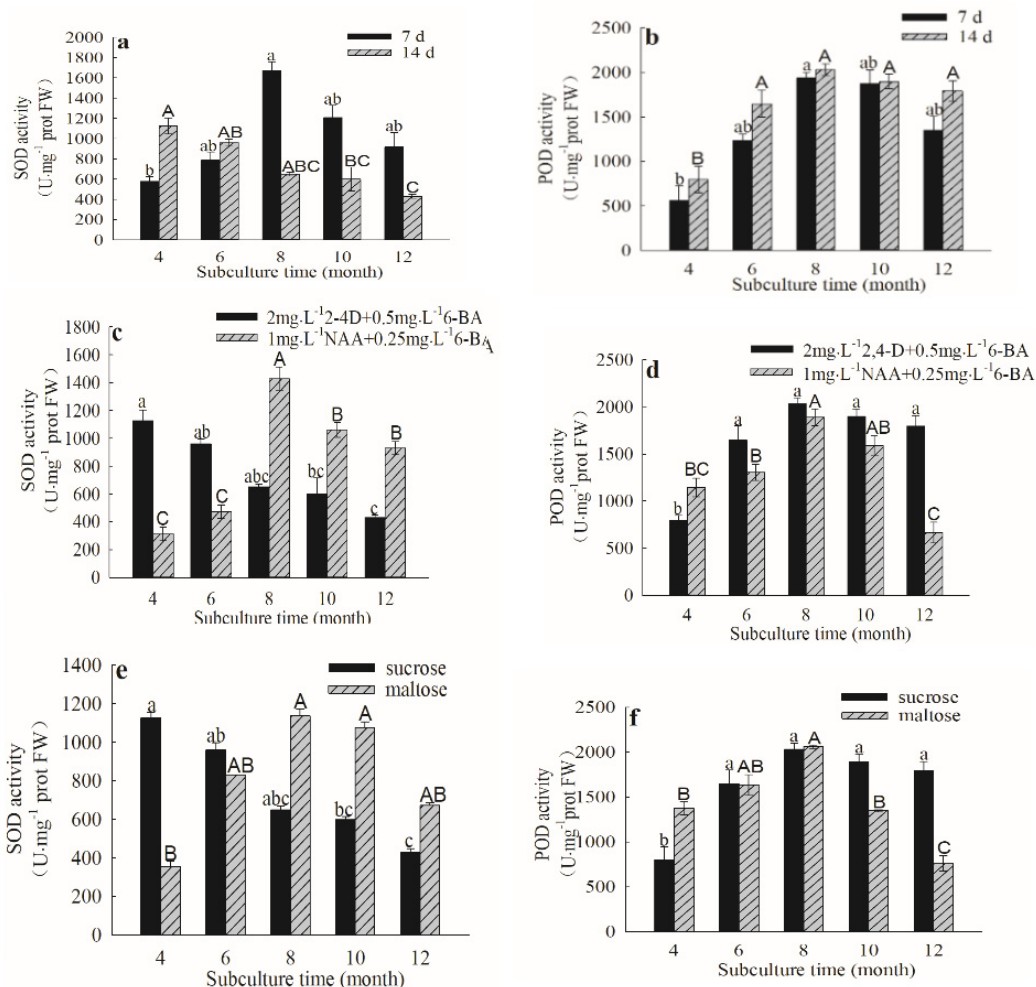

**Figure 6.** Effects of subculture cycle, hormone ratio, and carbon source on antioxidants in embryogenic callus of *Pinus koraiensis* during long-term subculture. (**a**) SOD activity of callus in different subculture cycles, (**b**) POD activity of callus in different subculture cycles (**c**) SOD activity in callus with different hormone ratios, (**d**) POD activity in callus with different hormone ratios, (**e**) SOD activity in callus of different carbon sources, (**f**) POD activity in callus of different carbon sources;Note: Different letters (upper or lower case) in the table indicate significant differences between columns (Duncan's multiple range test; $p = 0.05$).

Effect of Hormone Ratio on Antioxidant

SOD activity of embryogenic callus on the medium supplemented with 2 mg·L$^{-1}$ 2, 4−D + 0.5 mg·L$^{-1}$ 6-BA decreased gradually with increased subculture time. SOD activity in the 4th month of subculture was the maximum (1126.53 U·mg$^{-1}$ prot), while the minimum (429.06 U·mg$^{-1}$ prot) was observed in the 12th month. SOD activity of callus on the medium supplemented with 1 mg·L$^{-1}$ NAA + 0.25 mg·L$^{-1}$ 6-BA increased initially and then decreased with increased subculture time. SOD activity was the highest in the 8th month of subculture (1429.48 U·mg$^{-1}$ prot) (Figure 6c). Trends in POD activity, CAT activity, and H$_2$O$_2$ content of callus on the medium supplemented with 2 mg·L$^{-1}$ 2, 4−D + 0.5 mg·L$^{-1}$ 6-BA and 1 mg·L$^{-1}$ NAA + 0.25 mg·L$^{-1}$ 6-BA were consistent with increased subculture time, reaching the maximum values in the 8th month (Figure 6d, Figure S1c, online Supplementary Data) and the 12th month (Figure S1d, online Supplementary Data) of subculture, respectively.

Effect of Carbon Source on Antioxidants

On the medium with sucrose as the carbon source, SOD activity of embryogenic callus decreased gradually with increased subculture time. SOD activity was the maximum (1126.53 U·mg$^{-1}$ prot) in the 4th month of subculture. On the medium with maltose as the carbon source, SOD activity of callus increased initially and then decreased with increased subculture time. There was a significant difference in the SOD activity between the 12th month of subculture and the 4th month of subculture ($p < 0.05$). SOD activity was the lowest (356.12 U·mg$^{-1}$ prot) in the 4th month of subculture and reached the maximum (1138.09 U·mg$^{-1}$ prot) in the 8th month of subculture (Figure 6e). On the medium with sucrose and maltose as the carbon sources, CAT activity and POD activity of callus increased initially and then decreased with increased subculture time, reaching the maximum value in the 8th month of subculture (Figure 6f, Figure S1e, online Supplementary Data). On the medium with sucrose or maltose as the carbon source, $H_2O_2$ content of callus reached the maximum in the 12th month of subculture (Figure S1f, online Supplementary Data).

## 4. Discussion

### 4.1. Shorter Subculture cycles Is More Favorable to Embryogenic Maintenance of Long-Term Subculture of Calli

The plant culture cycle plays a key role in yields from plant somatic embryogenesis. In this study, the number of somatic embryos of embryogenic calli subcultured with subculture cycles of 7 days or 14 days was 0 in the 12th month, indicating that they had lost the capacity for somatic embryogenesis entirely. From the perspective of long-term subculture, the number of somatic embryos of embryogenic calli subcultured with a cycle of 7 days was greater than that with a cycle of 14 days from the 8th month, indicating that the 7-day subculture cycle was more suitable for long-term subculture of *P. koraiensis* embryogenic calli and maintained the embryogenesis of the calli. This phenomenon has been reported for other tree species. In a study of the long-term subculture of *P. pinaster*, subculture once a week effectively alleviated the senescence of embryogenic callus [3].

High IAA content indicates vigorous growth of plants and promotes somatic embryogenesis of embryogenic calli [16]. In this experiment, the IAA content of embryogenic callus after 8 months of subculture with a subculture cycle of 7 days was more than that with a subculture cycle of 14 days, confirming that high IAA content in subculture is beneficial to the vigorous growth of embryogenic calli and the high potential for somatic embryogenesis.

Soluble sugar provides indispensable energy as a carbon source for embryonic development [17]. In this experiment, the soluble sugar content of embryogenic calli subcultured with a subculture cycle of 7 days was higher than that with a subculture cycle of 14 days after eight months of subculture, indicating that 7 days of subculture was more conducive to energy storage and maintenance of the potential for somatic embryogenesis. Mckersie [18] suggested that during somatic embryogenesis, starch granules are usually deposited near areas with strong activity. In this experiment, the starch content of embryogenic calli decreased gradually. The starch content of calli subcultured with a subculture cycle of 7 days did not decrease significantly and was higher than that of calli subcultured with a subculture cycle of 14 days, indicating that a subculture cycle of 7 days was more conducive to long-term subculture of embryogenic calli of *P. koraiensis*.

Peng et al. showed that excessive $H_2O_2$ accumulation may lead to the loss of embryogenesis of callus cells [13]. In this experiment, after eight months (except for the 12th month) of subculture with a subculture cycle of 7 days, the $H_2O_2$ content of embryogenic callus was lower than that for 14 days, indicating that a subculture cycle of 7 days was more suitable for the long-term subculture of *P. koraiensis* embryogenic callus and the maintenance of somatic embryogenic ability.

### 4.2. Proper Hormone Ratio Affects Embryogenic Maintenance of Long-Term Subculture of Embryogenic Calli

Unlike zygotic embryos, somatic embryos are not the product of double fertilization leading to the formation of a zygote, and lack endosperm to provide nutrients and plant growth regulators [19]. During callus subculture, using NAA instead of 2, 4-D or reducing the concentration of 2, 4-D is conducive to the maturation of somatic embryos [8]. In this experiment, the number of somatic embryos on the medium supplemented with 1 mg·L$^{-1}$ NAA + 0.25 mg·L$^{-1}$ 6-BA was greater than that with 2 mg·L$^{-1}$ 2, 4-D + 0.5 mg·L$^{-1}$ 6-BA, except at the 6th month of subculture. The results showed that NAA instead of 2, 4-D in the proliferative medium was beneficial for the maintenance of somatic embryogenesis during the long-term subculture of callus. This phenomenon was also found for sugarcane (*Saccharum officinarum*) [20].

During the development of somatic embryos of *Daucus carota* [21], *Pinus taeda* [22], and other plants, the transition from embryonic cell mass to spherical embryo was accompanied by increased endogenous ABA content. In this experiment, the number of somatic embryos of embryogenic calli in the medium supplemented with two hormones was highest when ABA content was highest, indicating that ABA played an important role in regulating the growth and development of somatic embryos. In a study of somatic embryogenesis of *Schisandra chinensis* [23], an appropriate increase of IAA content was conducive to promoting the initiation of somatic embryogenesis. In this experiment, the IAA content of embryogenic callus on the medium supplemented with 1 mg·L$^{-1}$ NAA + 0.25 mg·L$^{-1}$ 6-BA was generally lower but the number of somatic embryos was higher than on the medium supplemented with 2 mg·L$^{-1}$ 2, 4-D + 0.5 mg·L$^{-1}$ 6-BA, indicating that an appropriate amount of endogenous IAA content was conducive to somatic embryogenesis.

### 4.3. Maltose as Carbon Source Is More Favorable to Embryogenic Maintenance of Long-Term Subculture of Embryogenic Calli

The carbon source in the medium plays an important role in conifer somatic embryogenesis. In a study on the hybrid *Liriodendron chinense*, sucrose as the carbon source in the culture medium did little damage to embryonic cells [24]. During carrot culture, Verma and Dougall found that when selecting the sole carbon source of the culture medium, many carbohydrates were comparable to sucrose in their impacts on growth and embryogenesis [25]. In a study on *Medicago sativa* [26], when 3% maltose was added to the culture medium, the amount of somatic embryo maturation was highest, which was similar to the results for *Theobroma cacao* [27]. In this experiment, except for the 4th and 12th months of subculture, the number of somatic embryos on the medium with maltose as the carbon source was greater than that on the medium with sucrose as the carbon source, which indicated that the medium with maltose as the carbon source was more suitable for maintaining the potential for somatic embryogenesis during long-term subculture of embryogenic callus, which was similar to the results for *Theobroma cacao* [27].

In a study on *Taxus wallichiana* [28], contents of soluble sugar, starch, and protein in embryogenic callus were higher than those in non-embryogenic callus, indicating that the accumulation of energy was the material and energy basis for the transformation of embryogenic callus into somatic embryos. Similar results were found for *Musa acuminata* [29] and *Medicago sativa* [30]. In this experiment, the content of soluble sugar, starch, and protein of embryogenic callus subcultured with maltose as the carbon source generally was higher than that with sucrose as the carbon source, which indicated that in long-term subculture of *P. koraiensis* embryogenic callus, the medium with maltose as the carbon source provided better energy and was more suitable for maintaining the embryogenic potential of embryogenic callus.

SOD is the first line of defense against reactive oxygen species [31] and can catalyze the disproportionation reaction of antioxidant anions, remove reactive oxygen species, prevent membrane lipid peroxidation, and have anti-aging effects [32]. In a study of *Crocus sativus*, the activities of SOD and CAT increased at the early stage of somatic embryo

development [33]. In this experiment, the activities of SOD and CAT of embryogenic callus were higher in the medium with maltose as the carbon source than in the medium with sucrose as the carbon source after eight months of subculture, which was conducive to the maintenance of somatic embryogenesis.

## 5. Conclusions

In summary, it was found that the shorter subculture cycle, adding 1 mg·L$^{-1}$ NAA + 0.25 mg·L$^{-1}$ 6-BA to the medium, and using maltose as the carbon source could be more favorable to embryogenic maintenance of long-term subculture of embryogenic calli. Stored materials (soluble sugar, starch, and protein) in embryogenic calli provided an energy source for the somatic embryogenesis of *P. koraiensis*. High contents of IAA, SOD, and CAT and low content of $H_2O_2$ were beneficial to the somatic embryogenesis of embryogenic callus. These results revealed the physiological mechanism of the loss of somatic embryogenic ability of embryogenic callus of *P. koraiensis*, providing information for the long-term maintenance of the somatic embryogenic ability of embryogenic callus of *P. koraiensis*.

**Supplementary Materials:** The following supporting information can be downloaded at: https://www.mdpi.com/article/10.3390/f13101557/s1, Figure S1: Effects of subculture cycle, hormone ratio, and carbon source on antioxidants in *Pinus koraiensis* embryogenic callus during long-term subculture.

**Author Contributions:** Conceptualization, L.Y. and H.S.; methodology, Y.R.; software, X.Y.; validation, Y.R., X.Y. and L.Y.; formal analysis, Y.R.; investigation, X.Y.; resources, Y.R.; data curation, Y.R. and H.X.; writing—original draft preparation, Y.R. and X.Y.; writing—review and editing, L.Y., H.S., I.N.T. and A.M.N.; visualization, Y.R., H.X. and L.Y.; supervision, H.S.; project administration, H.S.; funding acquisition, L.Y. All authors have read and agreed to the published version of the manuscript.

**Funding:** The work was supported by the Fundamental Research Funds for the Central Universities (2572020DR05); the Innovation Project of State Key Laboratory of Tree Genetics and Breeding (2021B01);the National Key R&D Program of China (2017YFD0600600).

**Data Availability Statement:** Not applicable.

**Conflicts of Interest:** The authors declare no conflict of interest.

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
