# Peer review of "Interaction of Subculture Cycle, Hormone Ratio, and Carbon Source Regulates Embryonic Differentiation of Somatic Cells in Pinus koraiensis"

_forests, doi:10.3390/f13101557_

Round 1
Reviewer 1 Report
This manuscript mainly introduced the effects of different subculture cycles, hormone ratios and carbon sources on somatic embryogenesis of Pinus koraiensis. By setting variables, the number of somatic embryos under different conditions was counted, and the relevant physiological indices were measured. However, there are some major issues to be addressed before the manuscript considered for publication. According to the results shown in figures, all of the experiments are single factor experiments. However, the authors claimed the conclusion in Abstract L25-27 that the best solution for somatic embryogenesis comprise four factors, which can only be reached according to multiple factor experiments. The pictures of callus under treatment and control were not shown in the results. The introduction and discussion lack in-depth thinking. Some of the experiments need to be measured by more accurate methods.
1. L30-31, “Further, low levels of endogenous abscisic acid (ABA) and high levels of IAA could inhibit somatic embryogenesis of P. koraiensis.” The effect of ABA and IAA on somatic embryogenesis can not be speculated merely based on the level of ABA and IAA in callus.
2. Part 2.5, ELISA is not accurate for IAA and ABA detection, HPLC is recommended.
3. Fig1, what are the hormone ratios and carbon sources used in 7d/14d subculture? Is it 7d/14d subculture with sucrose? Is it a single variable or a multi-variable experiment?
4.In this manuscript, there is plenty of data, but the figures and description of results are not incorporated to reach a conclusion.
5. Figure issues
(1) L25-27, “The somatic embryogenic potential of P. koraiensis was maintained for the longest duration with a 7-day subculture cycle of embryogenic callus, hormone concentrations of 1 mg·L-1 1-naphthaleneacetic acid (NAA) and 0.25 mg·L-1 6-benzyl amino-purine (6-BA), and maltose as the carbon source.”
Which FIG is this conclusion shown?
(2) L74 ,“morphology and physiological indices” Do you have morphological photos? Can the callus regenerate?
(3) The whole column has only two colors, but there are many indices.
6. Writing issue
(1)L34, “physiological mechanism” should be modified to “physiological features”
(2) L41, “Somatic embryogenesis is an important mechanism” This is an inappropriate description.
(3) L47-48,“For most plants, 2.4-dichlorophenoxy acetic acid (2, 4-D) is the auxin used to induce somatic embryos.” The introduction lacks in-depth thinking, other related hormones should also be introduced, as well as sugar, protein and peroxidase.
(4) L89-L97,make a title,“SE proliferation, maturation and capability evaluation”
(5) 2.2 2.3 2.4,modify it to “different subculture cycles, hormone ratios, and carbon sources set”
(6)2.5 modify it to “Chemical analysis of long-term subcultured embryogenic calli”
(7)L403-405,“High contents of IAA, SOD, and CAT and low content of H2O2 were beneficial to the somatic embryogenesis of embryogenic callus”. Are the levels of these indices the result or cause of somatic embryogenesis? Why is it beneficial to somatic embryogenesis?
(8) The Discussion subtitle is the same as the subtitle of the method, and should be changed to a one-sentence conclusion, with a sublimated summary of the results. These results are indeed consistent with those reported in other species, but the discussion should focus on the significance of these physiological indices.
Author Response
Response to Reviewer 1 Comments
Dear Reviewer,
Our sincere thanks to you for the time and effort that you have put into reviewing our manuscript! We found all the comments very constructive and helpful, and have revised our manuscript according to all comments. Please find, below, our point-by-point response to the comments raised.
Thank you for considering our revised manuscript!
Point 1. L30-31, “Further, low levels of endogenous abscisic acid (ABA) and high levels of IAA could inhibit somatic embryogenesis of P. koraiensis.” The effect of ABA and IAA on somatic embryogenesis can not be speculated merely based on the level of ABA and IAA in callus.
Response 1: This article mainly looks at the embryogenic ability of calli through the number of somatic embryos. ABA and IAA are only used as physiological indicators to reflect the possible embryogenic ability of calli.
Point 2. Part 2.5, ELISA is not accurate for IAA and ABA detection, HPLC is recommended.
Response 2: We will pay attention to the HPLC method in the future.
Point 3.Fig1, what are the hormone ratios and carbon sources used in 7d/14d subculture? Is it 7d/14d subculture with sucrose? Is it a single variable or a multi-variable experiment?
Response 3: Figure 1. The proportion of hormone used in the subculture of 7d / 14d was 2 mg·L-1 2,4-D+0.5 mg·L-1 6-BA and the carbon source was 0.088 mol·L-1 sucrose, which was a univariate experiment.
Point 4. In this manuscript, there is plenty of data, but the figures and description of results are not incorporated to reach a conclusion.
Response 4: We revised the conclusion.
Point 5.Figure issues
(1) L25-27,“The somatic embryogenic potential of P. koraiensis was maintained for the longest duration with a 7-day subculture cycle of embryogenic callus, hormone concentrations of 1 mg·L-1 1-naphthaleneacetic acid (NAA) and 0.25 mg·L-1 6-benzyl amino-purine (6-BA), and maltose as the carbon source.”
Which FIG is this conclusion shown?
Response 5(1): We modified the language expression.
Point 5 (2) L74 ,“morphology and physiological indices” Do you have morphological photos? Can the callus regenerate?
Response 5(2): No morphological photos were taken, but callus could proliferate and somatic embryogenesis could be maintained.
Point 5 (3) The whole column has only two colors, but there are many indices.
Response 5(3):In order to explain the figure more clearly.
Point 6.Writing issue.
- L34, “physiological mechanism” should be modified to “physiological features”
Response 6(1):The problems mentioned have been modified.
Point 6 (2) L41, “Somatic embryogenesis is an important mechanism” This is an inappropriate description.
Response 6(2):Modified as:Somatic embryogenesis is an important system of plant somatic genesis and genetic transformation, which has the potential of large-scale seedling breeding.
Point 6.(3) L47-48,“For most plants, 2.4-dichlorophenoxy acetic acid (2, 4-D) is the auxin used to induce somatic embryos.” The introduction lacks in-depth thinking, other related hormones should also be introduced, as well as sugar, protein and peroxidase.
Response 6(3):NAA is added in the text, because the experiment mainly wants to see the difference between 2,4-D and NAA.
Point 6 (4) L89-L97,make a title,“SE proliferation, maturation and capability evaluation”
Response 6(4):The text has been revised.
Point 6 (5) 2.2 2.3 2.4,modify it to “different subculture cycles, hormone ratios, and carbon sources set”
Response 6(5):The text has been revised.
Point 6 (6) 2.5 modify it to “Chemical analysis of long-term subcultured embryogenic calli”
Response 6(6):The text has been revised.
Point 6.(7) L403-405,“High contents of IAA, SOD, and CAT and low content of H2O2 were beneficial to the somatic embryogenesis of embryogenic callus”. Are the levels of these indices the result or cause of somatic embryogenesis? Why is it beneficial to somatic embryogenesis?
Response 6(7): L403-405, "high content of IAA, SOD and cat and low content of H2O2 are conducive to somatic embryogenesis of embryogenic callus". The level of these indicators is the reason for somatic embryogenesis, because these indicators are indicative in calli with high number of somatic embryos.
Point 6 (8): The Discussion subtitle is the same as the subtitle of the method, and should be changed to a one-sentence conclusion, with a sublimated summary of the results. These results are indeed consistent with those reported in other species, but the discussion should focus on the significance of these physiological indices.
Response 6(8): We modified the language expression in the discussion.

Reviewer 2 Report
Comments and Suggestions for Authors
The manuscript presented for review is focused on the determination of the conditions and technical requirements for maintaining the somatic embryogenic ability of Pinus koraiensis because off the established that in this species, during somatic embryogenesis the ability of embryogenic callus to produce embryos gradually decreases with long-term proliferative culture, and the ability of embryonic differentiation is lost after twelve months. In order to the establishment of a technical sys- tem to maintain the somatic embryogenic ability of embryogenic callus of P. koraiensis, different subculture cycles, hormone ratios, and carbon sources were investigated during the subculture of embryogenic callus induced from immature P. koraiensis embryos to maintain somatic embryogenic ability. In addition, morphology and physiological indices were observed to reveal the mechanism of the loss of somatic embryogenic ability of P. koraiensis embryogenic callus. During the propagation, the maintenance of somatic embryogenic ability eliminates repeated explant sampling and primary culturing and effectively improves the production efficiency of seedlings from tissue culture. In addition, long-term subculture of embryogenic callus and maintaining embryogenesis enables production of more somatic embryos and more regenerated plants. The results revealed the physiological mechanism of the loss of somatic embryogenic ability of embryogenic callus of P. koraiensis, providing information for the long-term maintenance of the somatic embryogenic ability of embryogenic callus. This study provides also a model for the maintenance of somatic embryogenesis of calli of other coniferous species.
The manuscript is constructed according to requirements of “Forests” and is well written and presented. The research methods applied are appropriate, comprehensive and sufficient to achieve the objectives of the study. The illustrative material is representative. The statistical analysis applied complement and support the presented results.
A continuation, some recommendations are given:
Abstract:
The first sentence “During somatic embryogenesis of Pinus koraiensis, the ability of embryogenic callus to produce embryos gradually decreases with long-term proliferative culture, which seriously affects large-scale application of this technology.” may by omitted, and the paragraph should start with: “In this study, the conditions and technical requirements for maintaining the somatic embryogenic ability of P. koraiensis embryogenic calli were determined. Embryogenic calli of the P. koraiensis 1-100 cell line were used to investigate conditions that maintain the embryonic differentiation ability of embryogenic callus. The somatic embryogenic potential of P. koraiensis … ”
Anyway, this paragraph needs some shortening - as it exceeds the 200 word maximum specified in the instructions to authors.
Keywords:
In my opinion, it would be better if they were presented in alphabetical order
Materials and Methods
- In line 90 a modified Litvay medium (MLV) is introduced, cited also in line 94, but is not specified the source in which this medium was described.
- In line 136:“ theseanalyses“ - a space is omitted
Discussion
- In the sentence in line 343-344:”Unlike zygotic embryos, somatic embryos do not produce oosperm and lack endosperm to provide nutrients and plant growth regulators.” the written that „somatic embryos do not produce oosperm“ is incorrect is incorrect from the point of view that the embryo cannot produce oosperm (zygote is the recommended term) because because it is the result of the division of the zygote. So, my suggestion for the cited sentence is as follows: “Unlike zygotic embryos, somatic embryos are not product of double fertilization leading to the formation of a zygote, and lack endosperm to provide nutrients and plant growth regulators.”
Author Response
Response to Reviewer 2 Comments
Dear Reviewer,
Our sincere thanks to you for the time and effort that you have put into reviewing our manuscript! We found all the comments very constructive and helpful, and have revised our manuscript according to all comments. Please find, below, our point-by-point response to the comments raised.
Thank you for considering our revised manuscript!
Point 1.Abstract:
The first sentence “During somatic embryogenesis of Pinus koraiensis, the ability of embryogenic callus to produce embryos gradually decreases with long-term proliferative culture, which seriously affects large-scale application of this technology.” may by omitted, and the paragraph should start with: “In this study, the conditions and technical requirements for maintaining the somatic embryogenic ability of P. koraiensis embryogenic calli were determined. Embryogenic calli of the P. koraiensis 1-100 cell line were used to investigate conditions that maintain the embryonic differentiation ability of embryogenic callus. The somatic embryogenic potential of P. koraiensis …”
Anyway, this paragraph needs some shortening- as it exceeds the 200 word maximum specified in the instructions to authors.
Response 1: All have been modified. Now the abstract have 185 words.
Point 2.Keywords:
In my opinion, it would be better if they were presented in alphabetical order
Response 2: The order has been adjusted.
Point3.Materials and Methods
-In line 90 a modified Litvay medium (MLV) is introduced, cited also in line 94, but is not specified the source in which this medium was described.
-In line 136:“ theseanalyses“ -a space is omitted
Response 3: Modified, the source of MIV medium is from reference 14.
Response: The space of "theseanalyses" has been added.
Point 4: Discussion
-In the sentence in line 343-344:”Unlike zygotic embryos, somatic embryos do not produce oosperm and lack endosperm to provide nutrients and plant growth regulators.”the written that, somatic embryos do not produce oosperm“is incorrect is incorrect from the point of view that the embryo cannot produce oosperm (zygote is the recommended term) because because it is the result of the division of the zygote. So, my suggestion for the cited sentence is as follows:“Unlike zygotic embryos, somatic embryos are not product of double fertilization leading to the formation of a zygote, and lack endosperm to provide nutrients and plant growth regulators.”
Response 4: This section has been modified to read:“Unlike zygotic embryos, somatic embryos are not product of double fertilization leading to the formation of a zygote, and lack endosperm to provide nutrients and plant growth regulators.”.
